# Efficient GPU-Accelerated Global Optimization for Inverse Problems

**Utkarsh**[1*]   **Vaibhav Kumar Dixit**[1*]   **Julian Samaroo**[1]   **Avik Pal**[1]   **Alan Edelman**[1]

**Christopher Rackauckas**[1,2,3]
[1]CSAIL MIT [2]JuliaHub Inc [3]Pumas-AI Inc
`{utkarsh5, vkdixit, jpsamaroo, avikpal, edelman, crackauc}@mit.edu`

## Abstract

This paper introduces a novel hybrid multi-start optimization strategy for solving inverse problems involving nonlinear dynamical systems and machine learning architectures, accelerated by GPU computing on both NVIDIA and AMD GPUs. The method combines Particle Swarm Optimization (PSO) and the Limited-memory Broyden–Fletcher–Goldfarb–Shanno (L-BFGS) algorithms to address the challenges in parameter estimation for nonlinear dynamical systems. This hybrid strategy aims to leverage the global search capability of PSO and the efficient local convergence of L-BFGS. We experimentally show faster convergence by a factor of up to $8 - 30\times$ in a few non-convex problems with loss landscapes characterized by multiple local minima, which can cause regular optimization approaches to fail.

## 1 Introduction

Inverse problem-solving in the realm of nonlinear differential equations plays a critical role across various scientific and engineering disciplines (Isakov, 2006). These problems typically involve deducing unknown parameter values that align a theoretical model with empirical data. A majority of applications of Scientific Machine Learning (SciML) involve solving such inverse problems, such as training of Neural Ordinary Differential Equations (ODEs) by Chen et al. (2018), Physics Informed Neural Networks by Raissi et al. (2019), and using machine learning augmented systems like Universal Differential Equations in Rackauckas et al. (2020). However, traditional optimization techniques encounter formidable challenges in these contexts, primarily due to the non-convex nature and the prevalence of numerous local minima (Ye et al., 2019; Isakov, 2006). This makes the robustness and performance of solving these problems a major bottleneck, and thus the current largest barrier to scaling SciML software (Krishnapriyan et al., 2021).

Optimization methods like Adam (Kingma & Ba, 2014) have become a staple in SciML inverse problems due to their efficacy in non-convex scenarios, coupled with fast derivative calculations. Their widespread adoption is facilitated by their integration into relevant software tools such as Neuromancer (Drgona et al., 2023) and DiffEqFlux.jl (Rackauckas et al., 2019). However, these methods were originally designed for large neural networks prevalent in tasks like Natural Language Processing (NLP) and Computer Vision Yao et al. (2021). In contrast, SciML often involves smaller neural networks but with frequent calls, such as within ODE solvers. Traditional scaling techniques, which rely on parallelizing large neural network calls, are less effective in SciML, where smaller architectures within physical models are common.

In order to overcome this issue, we rely on the recent results which demonstrate GPU-accelerated solving of differential equations which can be used to simulate thousands or millions of parameter sets simultaneously (Utkarsh et al., 2024). This allows one to effectively make use of GPUs to perform many objective function calls simultaneously. However, the gradient-based methods which are commonly used, such as gradient descent, Adam, or (L)-BFGS, require sequential calling of the objective function (Kingma & Ba, 2014; Liu & Nocedal, 1989). Therefore, to effectively make use of these new parallelization tools, alternative optimization strategies are required.

Modified strategies for between-objective call parallelization have been developed and demonstrated in literature before (Kucherenko & Sytsko, 2005; Tsoulos & Stavrakoudis, 2010). However, previous

---

*Equal contribution.

optimizers of this form were unable to make use of the differentiable programming, which has been found to greatly accelerate the training of SciML systems (Ma et al., 2021). To solve this gap, we leverage the known effectiveness of the GPU-accelerated (a)synchronous Parallel Swarm Optimization method (PSO) proposed by Kennedy & Eberhart (1995); Zhou & Tan (2009) to obtain points in neighborhoods of the local minima Schmitt (2015), which are then used to initialize a multi-start L-BFGS solves which converges at a super-linear rate to local optima. The proposed methodology thus brings together GPU-accelerated ODE solvers, differentiability of the ODE solvers, and global optimizers with between-objective call parallelization in order to achieve a scalable global optimization system on non-convex ODE inverse problems. The generated solvers use the `Optimization.jl` Dixit & Rackauckas (2023) and fully automate the GPU-acceleration and differentiation, meaning that many scientists can adopt these tools without any changes to their model code.

## 2 METHODOLOGY

### 2.1 ALGORITHM DESCRIPTION

PSO, an evolutionary computation technique (Kennedy & Eberhart, 1995), combines with L-BFGS, enhanced by GPU computing, to create a novel hybrid multi-start optimization method. PSO identifies initial points, leveraging L-BFGS for faster convergence. As PSO tends to converge to local optima (Schmitt, 2015), running L-BFGS iteratively on these points aids in finding the global minimum. This multi-start approach improves the likelihood of locating global minima in complex landscapes. GPU computing accelerates the process, enabling efficient handling of large-scale problems and datasets. See Algorithm 1 for details.

---

**Algorithm 1** Hybrid PSO-L-BFGS Optimization Framework

---

1: Initialize PSO with a population of particles
2: $w$ represents the inertia weight, $\phi_p$ and $\phi_g$ are cognitive and social coefficients, and $r_p, r_g$ are randomly generated numbers.
3: **while** iterations less than max iterations of PSO **do**
4:     **for** each particle in PSO **do**
      • Update the particle's velocity: $v_{i,d} \leftarrow wv_{i,d} + \phi_p r_p (p_{i,d} - x_{i,d}) + \phi_g r_g (g_d - x_{i,d})$
      • Update the particle's position: $x_i \leftarrow x_i + v_i$
5:     **end for**
6: **end while**
7: Identify promising regions from PSO
8: $g_k$ is the gradient, $H_k$ is the approximate inverse Hessian, $s_k$ and $y_k$ represent differences in positions and gradients, $\rho_k = 1/(y_k^T s_k)$, and $\alpha_k$ denotes the step size which is determined with line searching
9: **while** iterations less than max iterations of L-BFGS **do**
10:     **for** each particle in swarm **do**
      • Calculating the gradient: $g_k = \nabla f(x_k)$.
      • Updating the Hessian approximation using recent gradients: $H_k \approx (I - \rho_k s_k y_k^T) H_{k-1} (I - \rho_k y_k s_k^T) + \rho_k s_k s_k^T$.
      • Determining the descent direction: $p_k = -H_k g_k$.
      • Position update: $x_{k+1} = x_k + \alpha_k p_k$.
11:     **end for**
12: **end while**

---

### 2.2 GPU BASED PARALLELIZATION

Broadly, the parallelization of the PSO can be divided into three parallel steps. The stochastic updates to the position and velocity described in section 1 for all the particles are independent of each other, making them parallelizable. The cost function for multiple particles can be evaluated in parallel, and the global minima in every iteration can be computed using parallel prefix algorithm (Ladner & Fischer, 1980). Implementing these algorithms on GPUs requires writing kernels, which are often outside the expertise of scientists and practitioners who are less programming-savvy. Hence, we propose an automated GPU-acceleration pipeline extending composability with the rest of the SciML ecosystem. The algorithms work on multiple GPU backends and leverage CPU acceleration via abstractions written using `KernelAbstractions.jl` package (Churavy et al., 2023).

Variants of PSO algorithms exist which combine these operations together in order to reduce kernel launch and generation overheads, trading off speed and accuracy, and their performance is tied to the nature of the problem. We present parallelization of such various algorithms, such as:

- `ParallelSyncPSOKernel`: The basic PSO algorithm implementation. Requires synchronization of particles at every iteration to calculate the global best found yet.
- `ParallelPSOKernel(; global_update = true)`: A single kernel implementation of `ParallelPSOKernel` with queue-lock mechanism for efficient calculation of global update Wang et al. (2022).
- `ParallelPSOKernel(; global_update = false)`: An asynchronous version of PSO, where the particles independently evolve, only relying on their exploration. Generally the fastest algorithm of all, however, ends having the lowest loss reduction due to independent evolution.
- `HybridPSO`: An hybrid algorithm which combines the generic PSO algorithms with standard local optimizers. The algorithm sequentially parallelizes the PSO exploration and multi-start local search. It currently supports BFGS and L-BFGS as the local optimization method. Performs the best in terms of work per iteration until convergence, ensuring global optimality (see Section 3.1).

Moreover, we enable PSO methods to allow constraints using penalty functions (Parsopoulos et al., 2002). The process of GPU parallelization is made possible by static compilation via static arrays, and implementing non-allocating subroutines such as gradient calculation, and line search. We also parallelize the multi-start optimization scheme with L-BFGS on GPUs, where the initial guess is estimated from few iterations of the PSO particles. The algorithm essentially "solves" the optimization problem with different initial guesses in parallel inside a GPU, and returns the best candidate for the optimization solution. We also ensure that the gradient calculations within the GPU kernel are performed with reverse mode automatic differentiation.

## 3 EXPERIMENTS AND BENCHMARKS

To benchmark our algorithms, we use both CPUs and GPUs for parallelization, with setup described in Appendix A.1. Primarily in benchmarking, we only report the timings for solve times, albeit any cache allocations such as for particle initialization. The test-suite and code is available open-source in Julia (Bezanson et al., 2017) at https://github.com/SciML/PSOGPU.jl.

### 3.1 GPU VS CPU: COMPARISON FOR EFFICIENT PARALLELISM

We benchmark our implementations to compare against CPU multi-threading with single precision arithmetic. The test problem is the 10-dimensional version of the non-convex Rosenbrock problem A.2.1. Figure 1 a demonstrates that GPU parallelization becomes feasible for $> 100-1000$ particles where HybridPSO is $\approx 10\times$ faster and asynchronous $100\times$ (see appendix Table 2). Furthermore, instantiating more particles exposes parallelism from the algorithm, which can achieve scalability at high dimensional problems.

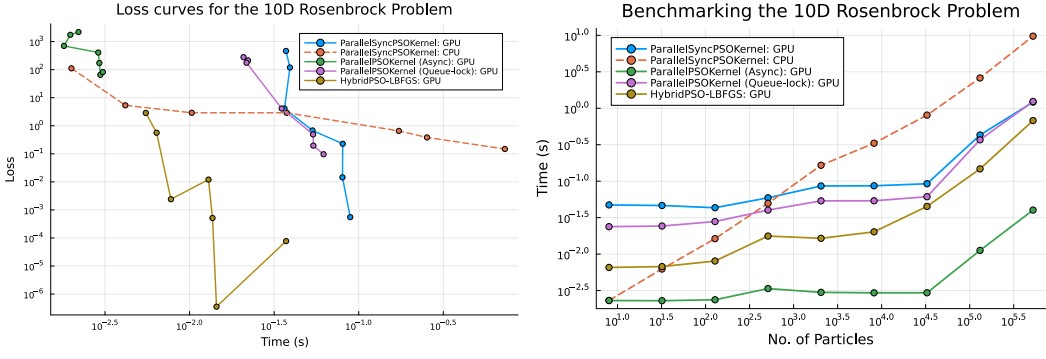

Figure 1: GPU parallelization becomes feasible, having at least $100-1000$ swarm size. HybridPSO-LBFGS has the best performance on loss vs time by varying number of particles.

### 3.2 Neural Ordinary Differential Equations as Inverse Problems

Neural ODEs proposed by Chen et al. (2018) are ML architectures popularly used in SciML. These are based on the Initial Value Problems (IVPs) ODEs where an initial condition $u(t_0)$ is given, and the solution is required to obtain in between the time interval $(t_0, t_f)$. Described as a continuous generalization of the Residual Network architecture in Chen et al. (2018), they are explicitly defined as $\frac{du(t)}{dt} = f_\theta(u(t), t)$, where $f_\theta$ is the neural network defined by the specific architecture. Efficient training of neural ODEs requires continuous or discrete adjoint methods. Using continuous adjoint results in constant memory overhead. However, it results in slower training due to the backward solving of an ODE (Chen et al., 2018). Trivially, training a Neural ODE can be seen as a parameter estimation problem for the ODE. We consider the fitting of the spiral ODE data (refer Section A.2.2 for further details) using a Neural ODE of relevantly small size. GPU-parallelization of small neural networks does not effectively overcome the overhead. Hence, we use `SimpleChains.jl`, a Julia package for faster training on CPUs. `SimpleChains.jl` makes neural networks amenable to GPU parallelization for multiple evaluations, and hence, using methods such as PSO, which offer GPU-parallelization of the optimizer instead of loss computation and gradients, can result in GPU utilization where traditional parallelization fails to prove beneficial. Figure 3 and Table 1 demonstrate that our algorithm performs better in terms of the accuracy of the results as well as the computational time by approximately $3\times - 8\times$. We also obtain similar results on ODE inverse problems, described in the Appendix A.3.

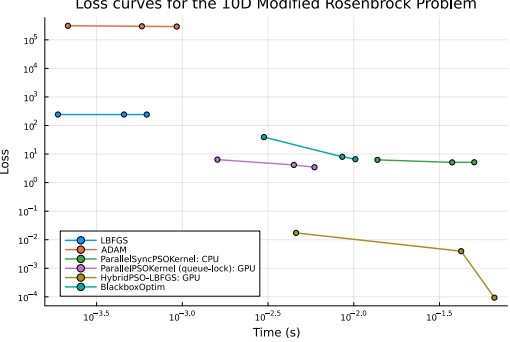

| Optimizer | Loss | Time (s) |
|-----------|--------|----------|
| Adam | 20.2722 | 1.4014 |
| L-BFGS | 12.6824 | 4.3721 |
| **GPU-PSO** | **0.5962** | **0.4826** |

Table 1: GPU-PSO performs the best on training small Neural ODEs.

Figure 2: In loss-time plots, HybridPSO-LBFGS excels, reaching the global minima, while LBFGS and Adam get trapped at local minima.

### 3.3 Failure of Existing Methods for Finding Global Optima

To demonstrate the effectiveness of HybridPSO methods, we conducted a comparison of their performance on the modified Rosenbrock problem (see Appendix A.2.1) across varying numbers of maximum iterations. As depicted in Figure 2, traditional gradient-based optimizers (Adam and L-BFGS) gets stuck at local minima (Mascarenhas, 2004), underscoring the necessity for global optimizers. Notably, HybridPSO-LBFGS achieves a reduction in loss by a factor of 1000, while maintaining computational efficiency comparable to that of the naive PSO and also with the standard differential evolution optimizer implemented in `BlackBoxOptim.jl` (Feldt, 2018).

## 4 Conclusion and Future Work

We demonstrated general purpose global optimization methods which allow for parallelization across objective function evaluations and uses differentiability to accelerate convergence beyond traditional techniques. The HybridPSO-LBFGS performs the best across our benchmarks, having the fast convergence performance of quasi-Newton (BFGS) type schemes while having the robustness to non-convexity indicative of other global optimization schemes. As such, it's a highly flexible method for difficult non-convex loss landscapes. The solvers are compatible with the `Optimization.jl` interface, ensuring accessibility for large scientists without the need for code adjustments.

However, there are a few caveats to this work. For one, it relies on the ability to have kernel generation for fast GPU-parallel calls to the objective function. As previously demonstrated, machine learning frameworks such as JAX (Bradbury et al., 2018) and PyTorch (Paszke et al., 2019) rely on array-based GPU parallelism, which have shown to be suboptimal in the context of ODE solvers (Utkarsh et al., 2024). As our inverse problem benchmarks heavily rely on these optimized kernels, the improved performance of the HybridPSO-LBFGS thus relies on the kernel generation approach and thus the method may not be replicable to other frameworks without major performance loss. In addition, this approach requires differentiability, i.e. compatibility with automatic differentiation, and thus it can restrict the types of functions a user may use within the loss function.

## ACKNOWLEDGMENTS

The authors acknowledge the MIT SuperCloud and Lincoln Laboratory Supercomputing Center for providing HPC resources that have contributed to the research results reported within this paper. This material is based upon work supported by the National Science Foundation under grant no. OAC1835443, grant no. SII-2029670, grant no. ECCS-2029670, grant no. OAC-2103804, and grant no. PHY-2021825. We also gratefully acknowledge the U.S. Agency for International Development through Penn State for grant no. S002283-USAID. The information, data, or work presented herein was funded in part by the Advanced Research Projects Agency-Energy (ARPA-E), U.S. Department of Energy, under Award Number DE-AR0001211 and DE-AR0001222. We also gratefully acknowledge the U.S. Agency for International Development through Penn State for grant no. S002283-USAID. The views and opinions of authors expressed herein do not necessarily state or reflect those of the United States Government or any agency thereof. This research was funded by DARPA under agreements HR00112090067. This material was supported by The Research Council of Norway and Equinor ASA through Research Council project "308817 - Digital wells for optimal production and drainage". Research was sponsored by the United States Air Force Research Laboratory and the United States Air Force Artificial Intelligence Accelerator and was accomplished under Cooperative Agreement Number FA8750-19-2-1000. The views and conclusions contained in this document are those of the authors and should not be interpreted as representing the official policies, either expressed or implied, of the United States Air Force or the U.S. Government. The U.S. Government is authorized to reproduce and distribute reprints for Government purposes notwithstanding any copyright notation herein.

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

## A  APPENDIX

### A.1  EXPERIMENTAL SETUP

The CPU benchmarks are performed on Intel(R) Xeon(R) CPU E5-2603 v4 @ 1.70GHz with 12 threads enabled. The GPU codes were executed using NVIDIA Tesla V100S-32 GB.

### A.2  PROBLEMS USED IN BENCHMARKING EVALUATIONS

#### A.2.1  ROSENBROCK FUNCTION

The Rosenbrock function used in GPU vs. CPU benchmarking is given as:

$$f(\boldsymbol{u}, \boldsymbol{p}) = \sum_{i=1}^{N-1} p_2(u_{i+1} - u_i^2)^2 + (p_1 - u_i)^2 \tag{1}$$

The modified $N$ dimensional Rosenbrock function used in our test-case is defined as:

$$f(\boldsymbol{u}, \boldsymbol{p}) = \sum_{i=1}^{N-1} p_2 sin(p_3 x_i)(u_{i+1} - u_i^2)^2 + (p_1 - u_i)^2 \tag{2}$$

Where, $N = 10$, $\boldsymbol{u} = [u_1, \ldots, u_N]^T$ and $\boldsymbol{p} = [p_1, p_2, p_3]^T = [1.0, 100.0, 1.5]^T$. The initial condition is $\boldsymbol{u_0} = \{5.0, \ \forall i \ \in 1 \ldots N\}$ The bounds assumed for the problem are $[-1.0, 5.0]$ on the optimization variables. The known global optima exists at $\boldsymbol{u^*} = \{1.0, \ \forall i \ \in 1 \ldots N\}$ with $f(\boldsymbol{u^*}, \boldsymbol{p}) = 0$.

#### A.2.2  SPIRAL DATASET FOR NEURAL ODES

The Spiral dataset adapted from Chen et al. (2018) from the evaluation of Neural ODEs is the dataset generated from the ODE given as:

| PSO-Method | Speed-up |
|---|---|
| ParallelSyncPSOKernel (CPU) | $1.0\times$ |
| ParallelSyncPSOKernel (GPU) | $3.4\times$ |
| ParallelSyncPSOKernel (Queue-Lock) | $4.2\times$ |
| HybridPSO-LBFGS (GPU) | $8.1\times$ |
| ParallelPSOKernel(Async) (CPU) | $102.8\times$ |

Table 2: Particle count affects average speed-up. Asynchronous version excels, but tends to achieve lower loss reduction overall.

$$\frac{du}{dt} = -0.1u^3 - 2.0v^3 \tag{3}$$

$$\frac{dv}{dt} = +2.0u^3 - 0.1v^3 \tag{4}$$

We generate the dataset with initial condition for the ODE as $[2.0, 0.0]^T$, where the integration is performed between $t \in [0.0\ s, 1.5\ s]$, generating 30 data-points, uniformly sampled in time. For benchmarking with PSO methods, we use $10,000$ particles for exploration, with 100 maximum iterations.

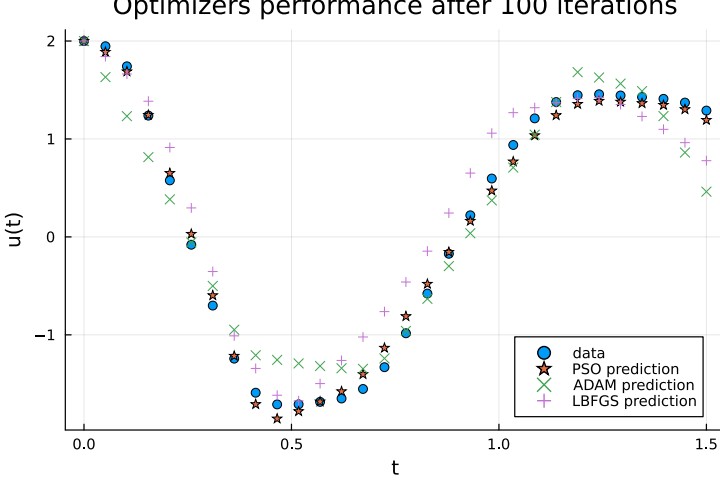

Figure 3: GPU-PSO methods outperform standard gradient-based optimizers in Neural ODEs.

### A.3 PARAMETER ESTIMATION BENCHMARKS

We benchmark some ODE parameter estimation problems to assess the tractability of GPU-enhanced PSO.

#### A.3.1 LOTKA VOLTERRA

The predator-prey "Lotka Volterra" equation is given as:

$$\frac{dx}{dt} = \alpha x + \beta xy \tag{5}$$

$$\frac{dy}{dt} = \delta y - \gamma xy \tag{6}$$

With initial condition as $[1.0, 1.0]$, and the integration being performed between $t \in [0.0\ s, 10.0\ s]$. We generate the dataset from this ODE to infer the parameters $\alpha, \beta, \gamma, \delta$.

| Optimizer (100 iterations) | Loss | Time (s) |
|---|---|---|
| Adam * | Fails | 50.3152 |
| L-BFGS * | Fails | 0.8967 |
| **GPU-PSO** | $\mathbf{1.5725 \times 10^{-7}}$ | **0.2303** |

Table 3: GPU-PSO performs the best on the parameter estimation of the Lotka-Volterra problem.

* Adam and L-BFGS exit with warnings as they result in unstable ODE solution.

### A.3.2 FITZHUGH-NAGUMO

The Fitzhugh-Nagumo model is a simplification of the Hodgkin-Huxley model (Izhikevich & FitzHugh, 2006). The dynamics is specified by the ODE given as:

$$\frac{dv}{dt} = v - \frac{v^3}{3} - w + l \tag{7}$$

$$\frac{dw}{dt} = \tau_{inv}(v + a - bw) \tag{8}$$

With initial condition as $[1.0, 1.0]$, and the integration being performed between $t \in [0.0\ s, 30.0\ s]$. We generate the dataset from this ODE to infer the parameters $a, b, \tau_{inv}, l$.

| Optimizer (100 iterations) | Loss | Time (s) |
|---|---|---|
| Adam | 0.0049 | 0.0204 |
| L-BFGS | 0.0015 | 0.5081 |
| **GPU-PSO** | $\mathbf{9.5449 \times 10^{-6}}$ | **0.1708** |

Table 4: GPU-PSO performs the best on the parameter estimation of the Fitzhugh-Nagumo problem.

