# OpenReview forum: "Efficient GPU-Accelerated Global Optimization for Inverse Problems"
_ICLR.cc/2024/Workshop/AI4DiffEqtnsInSci — AI4DiffEqtnsInSci @ ICLR 2024 Poster_

### Official Review · Reviewer_3dkH · 2024-02-15
**EFFICIENT GPU-ACCELERATED GLOBAL OPTIMIZATION FOR INVERSE PROBLEMS**

**Rating:** 7
**Confidence:** 3

**Review:**

The paper presents a commendable advancement in the domain of inverse problem-solving within nonlinear dynamical systems, integrating machine learning architectures and harnessing the computational power of GPU acceleration across both NVIDIA and AMD platforms. The proposed hybrid multi-start optimization strategy, which amalgamates Particle Swarm Optimization (PSO) and the Limited-memory Broyden–Fletcher–Goldfarb–Shanno (L-BFGS) algorithms, is an innovative approach that seeks to capitalize on the strengths of both global search capabilities and efficient local convergence properties.

The combination of PSO and L-BFGS for parameter estimation in nonlinear dynamical systems is particularly noteworthy, as it addresses a significant challenge in the field: the presence of non-convex problems with multiple local minima that can impede the effectiveness of conventional optimization methods. By leveraging the global search potential of PSO to navigate the broader parameter space and employing L-BFGS for its rapid local convergence, your strategy demonstrates a robust solution to overcoming the limitations posed by complex loss landscapes.

The reported acceleration factor of up to 8−30× in convergence speed is impressive and indicates a substantial improvement over existing methods. This acceleration not only showcases the efficiency of your hybrid strategy but also highlights the practical benefits of GPU computing in significantly reducing computation times for complex optimization problems.

---

### Official Review · Reviewer_pPjH · 2024-02-26
**EFFICIENT GPU-ACCELERATED GLOBAL OPTIMIZATION FOR INVERSE PROBLEMS**

**Rating:** 7
**Confidence:** 5

**Review:**

The authors demonstrate a "general purpose" GPU-accelerated global optimization method for inverse problems based on HybridPSO-LBFGS framework, which allows for parallelization across objective function evaluations and uses differentiability to accelerate convergence. The method is evaluated on a benchmark optimization test problem as well as to neuralODE training. The paper is well-written and the work is original and new based on the reviewer's knowledge of prior literature in this area of optimization methods. The proposed method can certainly speed up optimization problems encountered in SciML.

The authors should elaborate on the multi-start optimization scheme with L-BFGS and how it was parallelized on GPUs. On the other hand, it seems that the developed approach for HybridPSO-LBFGS is only going to be efficient in Julia. So, the method is not "general purpose" contrary to what the authors say in the Conclusion and Future Work section.

---

### Meta-Review · Area_Chair_X3Et · 2024-02-28

**Recommendation:** Accept (Poster)

**Metareview:**

Dear Authors,

Thank you for submitting the draft.

Both reviewers agree that the presented work presents interesting strengths. It is expected that authors will be addressing comments by the reviewers in the final draft.

regards

AC

---

### Decision · Program_Chairs · 2024-02-29

Accept (Poster)